# The Investigation of the Influence of a Cu_2_O Buffer Layer on Hole Transport Layers in MAPbI_3_-Based Perovskite Solar Cells

**DOI:** 10.3390/ma15228142

**Published:** 2022-11-17

**Authors:** Chunxiang Lin, Guilin Liu, Xi Xi, Lan Wang, Qiqi Wang, Qiyan Sun, Mingxi Li, Bingjie Zhu, David Perez de Lara, Huachao Zai

**Affiliations:** 1School of Science, Jiangnan University, Wuxi 214122, China; 2School of Internet of Things, Jiangnan University, Wuxi 214122, China; 3Wuxi Institution of Supervision & Testing on Product Quality, Wuxi 214101, China; 4Zhejiang Beyondsun Green Energy Technology Co., Ltd., Huzhou 313200, China; 5School of Materials Science and Technology, China University of Geosciences (Beijing), Beijing 100083, China

**Keywords:** perovskite solar cells, Cu_2_O, energy level alignment, carrier accumulation, SCAPS-1D

## Abstract

The passivation engineering of the hole transport layer in perovskite solar cells (PSCs) has significantly decreased carrier accumulation and open circuit voltage (*V_oc_*) loss, as well as energy band mismatching, thus achieving the goal of high-power conversion efficiency. However, most devices incorporating organic/inorganic buffer layers suffer from poor stability and low efficiency. In this article, we have proposed an inorganic buffer layer of Cu_2_O, which has achieved high efficiency on lower work function metals and various frequently used hole transport layers (HTLs). Once the Cu_2_O buffer layer was applied to modify the Cu/PTAA interface, the device exhibited a high *V_oc_* of 1.20 V, a high *FF* of 75.92%, and an enhanced *PCE* of 22.49% versus a *V_oc_* of 1.12 V, *FF* of 69.16%, and PCE of 18.99% from the (PTAA/Cu) n-i-p structure. Our simulation showed that the application of a Cu_2_O buffer layer improved the interfacial contact and energy alignment, promoting the carrier transportation and reducing the charge accumulation. Furthermore, we optimized the combinations of the thicknesses of the Cu_2_O, the absorber layer, and PTAA to obtain the best performance for Cu-based perovskite solar cells. Eventually, we explored the effect of the defect density between the HTL/absorber interface and the absorber/ETL interface on the device and recommended the appropriate reference defect density for experimental research. This work provides guidance for improving the experimental efficiency and reducing the cost of perovskite solar cells.

## 1. Introduction

Organic-inorganic perovskite solar cells (PSCs) have exhibited an exciting tendency, with high performance from 3.8% to nearly 26.0% in power conversion efficiency (PCE) [1], motivating great research interest in the field of photovoltaics. Although PSCs can achieve higher energy conversion efficiency, there is a long way to go regarding its theoretical limitation of PCE. One apparent reason is the charge accumulation between the metal electrode and functional layers. With the charge accumulated at the interface, the energy level mismatch and potential barrier effect were enhanced because of the reversed electric-field attached by the charge accumulation. Therefore, optimizing interfacial contact and improving interfacial energy level alignment is an important way to reduce charge accumulation and promote hole extraction, which has drawn the attention of the scientific community regarding PSCs.

The performance of an optoelectronic device critically depends on the transport ability of carriers between interfaces inside of PSCs. Charge accumulation or recombination would occur if the charge carriers encounter a higher barrier or a high interface-state density [2]. To address this problem, buffer layers have frequently been used to enhance carrier injection, decrease trap-states, and decrease contact resistance in various solar cells [2]. Specifically, the buffer layer between the hole transport layer (HTL) and the anode has a main function of blocking the migration of electrons towards the metal electrode, reducing interfacial recombination, and promoting hole transportation [3]. Several attempts have been made to modify the HTL/electrode in PSCs using organic interlayers to improve the interface and interlayer properties to overcome the interface loss. For instance, Arora et al. concluded that the instability of PSCs mainly originates from the CuSCN/Au interface contact and is not related to the degradation of the CuSCN/perovskite interface, so they introduced a thin conductive rGO (reduced graphene oxide) modified layer between CuSCN and Au to effectively alleviate the problem of interface degradation [4]. Zhou et al. introduced a poly-TPD ultra-thin layer to modify the HTL/Ag interface, thereby improving the efficiency of the solar cell (from 5.95% to 10.36%) because it can smooth the barrier between the HTL and electrode to promote the transmission of holes and passivate the surface defects to improve the interface contact [5]. However, for organic materials, the stability is a deadly point when they are employed in a complex environment. Therefore, researchers are exploiting transition metal oxides (TMOs) as interfacial buffer layers for extracting photogenerated holes; the metal oxides were demonstrated to exhibit good processability, transparency, and charge transport properties, as well as excellent stability [6]. Cai et al. found that the addition of NiO nanoparticles (as both the HTL and buffer layer) between the perovskite film and the carbon electrode can effectively promote the separation and extraction of photogenerated carriers and inhibit the charge recombination at the perovskite layer/carbon electrode interface, and they achieved the highest efficiency of 13.6% [7]. Zhao et al. demonstrated that the efficiency (11.4%) of a perovskite solar cell obtained by adding MoO_x_ between the HTL and Al electrode is comparable to that of a cell using a standard Ag top electrode, which is due to the effectiveness of the hole extraction using MoO_x_ [8]. On the downside, NiO_x_ is often grown by sputtering, which may destroy the organic charge transport and perovskite layer [9]. MoOx reacts strongly with the lead halide perovskite, hampering its long-term stability [10,11]. Nevertheless, NiO and MoO_x_ both show obvious disadvantages that limit the improvement of their efficiency because of their lower hole mobility of about 1.6 × 10^−4^–1.6 × 10^−3^ cm^2^/(V·s) and 7.8 cm^2^/(V·s), respectively [3]; thus, there is still significant room for exploration concerning HTL/electrode interface optimization. Overall, these issues continue to motivate further research to explore efficient buffer layers with relative stability and good interfacial contact. A recent theoretical study proposed that Cu_2_O might outperform other TMOs [12]. Moreover, Cu_2_O has been studied for decades as a typical p-type semiconductor due to its unique physical properties and applications in areas ranging from photo electrochemistry to magneto electrics and superconductors [13]. Apart from its natural p-type conductivity, Cu_2_O also possesses a high carrier mobility of over 100 cm^2^/(V·s) and a long carrier diffusion length ranging up to several micrometers [14]. Moreover, Cu_2_O can be grown at temperatures below 200 °C through soft growth methods using chemical vapor deposition (CVD) or atomic layer deposition (ALD) [15]. The unique characteristics of Cu_2_O make it a promising candidate for solar cell applications, and rare investigations have been carried out using Cu_2_O as a buffer layer to modify the HTL/electrode interface.

Thus, in this paper, we reported Cu_2_O as buffer between metal electrode/hole transport materials (HTMs) based on using a solar cell capacitance simulator (SCAPS-1D) to help understand the beneficial effect of Cu_2_O on device performance. The basic model structure of PSC we built is: metal electrode/Cu_2_O (with/without buffer layer)/HTL (multiple hole transport materials in model)/MAPbI_3_(perovskite absorber layer)/SnO_2_ (electron transport layer)/FTO (transparent conductive oxide). Initially, we employed Spiro-OMeTAD as the HTM and investigated the effect of Cu_2_O on different metal electrodes and found that Cu_2_O can effectively improve the performance of Cu-based devices. Then we compared the performance of the device, with and without a Cu_2_O buffer layer, based on a Cu electrode to vary different HTM, and the results showed that the device with PTAA-HTL showed the most enhanced performance (without Cu_2_O—PCE:18.99%, with Cu_2_O—PCE: 22.49%). Finally, we explained the optimization of the thickness combinations of the buffer layer for achieving the highest *PCE*.

## 2. Device Structure and Simulation Parameters

In this work, we use the key platform SCAPS-1D to simulate the MAPbI_3_-based heterojunction perovskite solar cells. The software SCAPS (version 3.3.10) is a superb and powerful tool for helping us to clearly understand the physical behavior of the different optoelectronic properties of any solar cell [16]. The principle of SCAPS software is based mainly on two basic equations: the Poisson equation and the continuity equation of electrons and holes under a steady-state condition.

The Poisson equation can be given by:(1)$$\frac{d^{2}}{dx^{2}}\psi \left(x\right)=\frac{e}{\varepsilon_{0}\varepsilon_{r}}\left(p\left(x\right)-n\left(x\right)+N_{D}-N_{A}+\rho_{P}-\rho_{N}\right)$$
where *ψ* is electrostatic potential, $\varepsilon_{0}$ and *ε_r_* are relative and the vacuum permittivity, *e* is electrical charge, *p* and *n* are hole and electron concentrations, *N_D_* and *N_A_* are charged impurities of the donor and acceptor type, and *ρ_P_* and *ρ_N_* are hole and electron distribution, respectively.

The continuity equations of electrons and holes can be described by:(2)$$\frac{dJ_{n}}{dx}=G-R$$
(3)$$\frac{dJ_{p}}{dx}=G-R$$
where *J_n_* and *J_p_* are electron and hole current densities, *G* is the generation rate, and *R* is the recombination rate.

In this simulation work, the structure of the solar cell model is: metal electrode/Cu_2_O (with/without buffer layer)/Spiro-OMeTAD (HTL)/MAPbI_3_ (perovskite absorber layer)/SnO_2_ (ETL)/FTO (transparent conductive oxide), as shown in Figure 1. The parameters (thickness, band gap, dielectric permittivity, electron affinity, electron/hole mobility, electron/hole thermal velocity, defect density, etc.) used for the solar cell structure in this simulation are shown in Table 1. Values of the input parameters are taken from the references given in the tables. The parameters of the interface defect layers are given in Appendix A. The parameters of the back and front electrodes are given in Appendix A (the work function of FTO set to 4.4 eV). The list of work functions of the back metal electrodes used are given in Appendix A.

In SCAPS, the data for the absorption coefficient (α) versus the wavelength (λ) can be imported from an external source; the absorption data used in this paper are taken from the literature, as shown in the Appendix A. The simulation is carried out under the AM 1.5G solar spectrum, with an incident power density of 1000 W/m^2^ at room temperature (300 K).

## 3. Result and Discussion

### 3.1. Reference Device Performance

Solar cell harvest photons and then convert energy to electric energy for output. Generally, the complex energy conversion process can be approximately summarized into three steps: (1) the photon absorption and electron excitation [23], in which the perovskite material absorbs the incident photons with energy greater than its band gap, and is excited to generate excitons; (2) the charge transportation [24], in which the excitons separate and transfer to the hole transport layer and the electron transport layer, respectively; (3) the charge extraction [25], in which the carriers drift to the electrodes and eventually form a current through the external circuit. Note that the recombination process is also an important step not mentioned here. Then, based on the above-mentioned conversion process, the selections of an absorber, transportation material, and electrodes, as well as the structural design, are crucial to PSCs. Methyl lead triiodide (MAPbI_3_) was applied in the model as the absorber layer because of its high absorption coefficient in the visible range, leading to excellent photoelectronic properties [26]. According to previous research [27], the thickness design was set in the 0.5 to 0.6 µm range for balancing the light absorption and charge extraction. Meanwhile, to simplify the model, we have chosen the most-used SnO_2_ as the ETL in this article because of its unique advantage in high electron extraction capability compared to other ETLs [28]. After referring to other works relating to SnO_2_ ETL, we determined that the device shows better performance when the SnO_2_ thickness is 0.07 µm, which inspired our model [20,29]. Simultaneously, the application of HTL is equally important to the device performance; the selection and the thickness optimization of the HTL are also hot research topics in the scientific community [30,31]. Particularly when using SnO_2_ as the ETL, most articles have found that the best thickness of the HTLs was around 0.15 µm. Finally, the metal electrode used to extract the charge also has a great influence on the solar performance. Based on the short summary above, MAPI_3_ perovskite devices based on SnO_2_ (ETL), Spiro-OMeTAD (HTL), and Ag (electrode) were initially applied in our model to check the simulation accuracy. The PCE of the simulated model was 20.13%, which is in accordance with Zhao’s report [32].

### 3.2. Effect of Cu_2_O Buffer Layer on Various Metal Electrodes

It was found that in HTL-free perovskite solar cells, iodide ions and methylamine ions at the grain boundaries easily diffuse and react with the Ag electrode surface, which will degrade the performance of the device [33]. Thus, to improve device stability, there is an urgent need to find materials that can replace Ag electrodes. With the proposed model, we can explore the effect of Cu_2_O on different metal electrodes by comparing the two cases, with or without the Cu_2_O buffer layer, to find the metal electrode that best matches Cu_2_O. Generally, Cr (4.5 eV), Cu (4.65 eV), Ag (4.7 eV), Au (5.1 eV), Ni (5.15 eV), and Pt (5.65 eV) (Appendix A) are the most frequently used electrode materials.

First, we clearly observed changes in the *Jsc*, *V_oc_*, *FF*, and *PCE* values by changing the work function of the back electrode without adding a Cu_2_O buffer layer. *Jsc* was almost unchanged, and *V_oc_* showed a brief promotion with a decreasing work function, then remained almost the same for a work function greater than 4.65 eV (work function of Au electrode) (Figure 2a,b). However, for both *FF* and *PCE*, the work function increased significantly and then reached a maximum at 5.1 eV (work function of Au electrode) (Figure 2c,d). These conditions are caused by the Schottky barrier between the HTL/electrode, which is inversely proportional to the back electrode work function, and the high energy barrier hinders the carrier transportation. For p-type semiconductors, a good ohmic contact is easily formed with a high metal work function [34]. Therefore, under the typical n-i-p structure, to obtain sufficient photovoltaic performance, it is required to select an expensive metal (such as Au), with a work function higher than 5.1 eV, for the back electrode, which is not conducive to the industrialization of perovskites.

After inserting the Cu_2_O buffer layer between the HTL/electrode, most of the parameters showed an increasing trend at the beginning, maintaining stability after 5.1 eV, and the change of *J_sc_* is only slight (Figure 3). However, by comparing the *FF*, *PCE* before and after inserting the Cu_2_O (Figure 3c,d, it can be clearly seen that the Cu_2_O buffer layer can effectively improve the *FF* of Cu electrode devices (from 66.07% to 77.03%), thus improving its *PCE* (from 18.92% to 21.46%), even for other devices with lower work function electrodes (Cr, Ag), the buffer layer can also have a similar effect. The Schottky barrier will form between the HTL and the low work function metal electrode, and the Schottky contact may lead to the decrease in *V_oc_* of the device [35]; this is the reason why the Cu electrode device is not efficient when the Cu_2_O buffer layer is not added. In similar research, Lin et al. added a CuO_x_ interlayer between the carrier transport layer and Ag or Al and found that the CuO_x_ film, which mainly consists of Cu_2_O, can effectively reduce the barrier height between this interface, transforming the Schottky contact into an excellent ohmic contact [36]. Moreover, Table 2 shows a photovoltaic parameters table showing different HTL devices, with and without a Cu_2_O buffer layer, using an Au electrode, and our simulated device results are compared with recently published reports. We can see that for these HTMs with wide bandgap (Table 2), the addition of a Cu_2_O buffer layer did not change the photovoltaic parameters (*J_sc_*, *V_oc_*, *FF*, *PCE*) while using an Au electrode device. For the remaining HTMs with a narrow bandgap, the addition of Cu_2_O only marginally improved the device performance. The few slightly increased PCEs may be attributed to the high hole mobility and wide band gap of Cu_2_O. Thus, we speculated that the addition of Cu_2_O can effectively reduce the Schottky barrier between the HTL and Cu, as well as the low work function of the metal, and facilitate hole transportation from the HTL to the electrode. However, for metals such as Au, in which the energy barrier between the high work function electrode is already small, the role of Cu_2_O in reducing the energy barrier appears to be minimal. Furthermore, according to previous research, the introduction of a trace amount of AgI at the HTL/Ag interface can effectively increase the work function of Ag, which eliminates the downward band bending between the HTL and the Ag electrode [37]. From another point of view, inserting a Cu_2_O buffer layer between HTL/Cu may increase the work function of the Cu electrode in a disguised form. Nevertheless, for an Au electrode, if the Cu_2_O also increases its work function, the performance of the device will not increase because all *PV* parameters have approached the maximum saturation value at 5.1 eV.

Therefore, inserting a Cu_2_O buffer between the HTL/electrode interface is an excellent strategy to improve solar cell performance with a Cu electrode. At the same time, J Huang et al. found that Cu metal was stable compared to other metals when MAPbI_3_ is used as the absorber layer [42]. As a low-cost and stable electrode, the application of Cu is an idea candidate for simulation, as well as further industrialization. In the following section, we will demonstrate the superiority of the Cu_2_O buffer layer over different HTLs.

### 3.3. The Effect of a Cu_2_O Buffer Layer with HTLs

To enhance the performance of perovskite solar cells, the HTL has a crucial impact on the device PCE by transporting holes and suppressing recombination after exciton dissociation [43]. The existing commercial hole transport material Spiro-OMeTAD exhibits disadvantages such as long synthesis cycle, complicated processing, high cost, and low stability, and the photoelectric conversion efficiency (*PCE)* of PSCs with Spiro-OMeTAD basically reaches the upper limit of 24.8% [44]. Therefore, finding low-cost and stable HTMs is urgently needed to realize the large-scale practical applications of PSCs.

Next, different HTMs (CuI, CuSCN, CuSbSe_2_, Spiro-OMeTAD, PTAA, PEDOT: PSS, P3HT) were used to optimize the simulation and guide the experimental research. The input parameters of the mentioned HTLs are listed in Table 3. The obtained *J–V* characteristic curves (with and without Cu_2_O) for all mentioned HTL devices are presented in Figure 4 and Appendix A. The photovoltaic parameters obtained using different HTL-based devices are shown in Table 4. We found that for most HTLs we used, the *J–V* curve of the reference device exhibited an S-shape around the *V*_oc_ site, resulting in low *FF*. Meanwhile, once the Cu_2_O buffer layer was inserted, the S-shape *J–V* curve was rapidly shifted. The modification of the S-shape was mainly caused by the energy alignment of Cu_2_O. The pure contact between HTL and Cu generates a stronger Schottky barrier, leading to a poor charge transportation. The Schottky barrier accelerated the charge accumulation, and the rest of the trapped carriers formed a reversed electric field against the built-in electric field; then the S-shaped *J–V* curve was formed due to the poor charge transportation [45]. Hence, the introduction of an additional buffer layer was essential to promote energy level alignment between the perovskite/HTL and HTL/Cu interfaces [46]. To further correct the mismatching, Cu_2_O is an idea candidate, as it exhibits a higher band gap to match between the HTL/Cu.

After testing several different HTMs using the structure of this work, we noticed that the PTAA-HTL based device exhibited a remarkable *PCE* of 22.49% greater than the other HTMs-based devices. Compared to the Spiro-OMeTAD-HTL device (*V_oc_* = 1.18 V, *J_sc_* = 23.55 mA/cm^2^), we obtained a higher *V_oc_*, *J_sc_* (1.20 V and 24.64 mA/cm^2^). The PTAA-HTM has the deepest valence band energy level with respect to MAPbI_3_ and Cu_2_O when compared with other HTMs (Figure 5), which allowed for the maximization of the *V_oc_*, and the large optical band gap (*E_g_* = 2.95 eV) of PTAA guarantees high hole blocking properties to prevent the transfer of photogenerated charges from perovskite to itself. The valence band maximum (*VBM*) of MAPbI_3_, PTAA-HTL, and Cu_2_O buffer layer are 5.45 eV, 5.25 eV, and 5.37 eV, respectively. Therefore, the energy-level offset (*ΔE*) for charge transfer between the *VBM* of MAPbI_3_ perovskite and PTAA-HTL is 0.20 eV, and the energy-level offset (*ΔE*) between the *VBM* of PTAA-HTL and the buffer is 0.12 eV. Additionally, the *V_oc_* was also increased (without Cu_2_O: *V_oc_* = 1.12 V, with Cu_2_O: *V_oc_* = 1.20 V), perhaps due to the decrease of recombination [52]. The wide gap band of Cu_2_O reduces recombination losses and pulls holes from the absorber, allowing for smoother extraction of the holes while experiencing lower resistance [53]. Meanwhile, Albert et al. found that the deposition of CuOx does not degrade the surface of perovskite or introduce traps at the interface by adding a layer of CuOx between the PTAA and the anode [54].

The CuSCN-HTL and CuI-HTL devices have the second-highest and third-highest efficiency of 21.70% and 21.58% (Table 4), respectively; here, the *VBM* of CuSCN and CuI are equivalent to that of Spiro-OMeTAD (Figure 5), but they have better hole mobility than Spiro-OMeTAD (Table 1, Table 4). Therefore, the addition of the Cu_2_O buffer layer has even balanced the *VBM* of CuSCN-HTL and CuI-HTL, which enhances the device performance after replacing the Spiro-OMeTAD (the *PCE* of the Cu_2_O/Spiro-OMeTAD device is 21.46%), It is worth noting that the addition of the Cu_2_O buffer layer has a simultaneous “slight” S-bend elimination effect on the PEDOT: PSS-HTL and P3HT-HTL devices, while it negatively affects CuSbSe_2_ (Appendix A). These phenomena can be attributed to their small band gaps. Compared to PTAA, despite having a close *VBM*, their lower conduction band minimum (*CBM*) leads to electron leakage from the perovskite absorber layer to the anode. The higher *CBM* of Cu_2_O helps to prevent electron leakage from the perovskite to the metal electrode, which may reduce the recombination of the HTL, the electrode, and the related interfaces, thereby reducing unnecessary *V_oc_* loss.

Consequently, compared to other HTMs, since PTAA has the deepest *VBM* and a suitable bandgap, the Cu_2_O material can achieve the best energy alignment with PTAA and MAPbI_3_, which makes the Cu_2_O well suited as a buffer layer in n-i-p structured perovskite devices [55].

### 3.4. The Optimization of Cu_2_O, Absorber Layer, and PTAA Thickness

According to the previous research shown in Section 3.3, based on the most effective structure of Cu/Cu_2_O/PTAA, the optimized thickness combinations were still absent in our study. Hence, first, we kept the thickness of PTAA-HTL at 0.15 µm, the tunable thickness of the Cu_2_O buffer layer (from 0.01 µm to 0.10 µm) and the absorber layer (from 0.40 µm to 1.0 µm), and *J_sc_*, *V_oc_*, *FF* and *PCE* were carried out; all results are summarized in Figure 6.

As shown in Figure 6a, the *J_sc_* changed slightly with the change in thickness of Cu_2_O, probably owning to the fact that the use of a thinner Cu_2_O will not affect the photo absorption. The photocurrent decreased from 25.45 mA/cm^2^ to around 24.20 mA/cm^2^ as the thickness of MAPbI_3_ decreased because the thinner absorption layer absorbs fewer photons, resulting in fewer electron-hole pairs. Conversely, the *V_oc_* shows an increasing tendency when the thickness of MAPbI_3_ and Cu_2_O decreases (Figure 6b), which may be due to the reduced recombination of free charge carriers in thinner function layers. We find that FF (Figure 6c) decreases with the decreasing thickness of Cu_2_O, which may be due to the fact that thinner Cu_2_O is less effective in improving charge accumulation. However, it increases with the decreasing thickness of MAPbI_3_ because thinner MAPbI_3_ results in lower series resistance of the device. Combined with the changes of *J_sc_* and *V_oc_*, it can be seen from Figure 6d that PCE increases with the decrease in Cu_2_O thickness until it is close to 0.02 µm, and it shows a clear trend of first rising and then falling with the decrease inMAPbI_3_ thickness, reaching the maximum value at 0.60 µm.

Based on the previous research, the optimized buffer layer thickness and absorber layer thickness are kept constant to obtain the optimal thickness of the PTAA. We changed the thickness of the PTAA from 0.01 µm to 1.0 µm (Figure 7), and we see that Jsc remains almost unchanged when the PTAA thickness is greater than 0.02 µm; Voc increases slowly as the PTAA thickness increases. PCE and FF decrease continuously when the PTAA thickness is greater than 0.02 µm, which may be due to higher series resistance with increasing PTAA thickness. It was found that the maximum PCE of the device was 22.94% when the thickness of PTAA was 0.02 µm.

Through our optimization, the device can achieve the best performance when the thickness of Cu_2_O, MAPbI_3_, and PTAA are 0.02, 0.60, and 0.02 µm, respectively, and the maximum *PCE* is 22.94%, *FF* is 77.20%, *J_sc_* is 24.93 mA/cm^2^, and *V_oc_* is 1.19V.

### 3.5. The Effect of Interface Layer Defect

The impact of the defect states in both interfaces in the proposed structure of the cell, the HTL/absorber, and the absorber/ETL have been investigated in detail. The defect parameters of the two interface layers are shown in Appendix A.

The influence of the defect density at the PTAA/absorber interface layer on the solar cell parameters was changed from 10^9^ cm^−2^ to 10^13^ cm^−2^, while the other variables remained unchanged. The results we obtained are as shown in Figure 8. We can see that the increase in the defect density results in a decrease in the *V_oc_*, *J_sc_* and *PCE*, while the *FF* shows an upward trend. In regards to *FF* as the ratio of maximum power area to the product of *J_sc_* and *V_oc_* in the *J–V* curve, its increase may be attributed to a slightly unchanged maximum power point (MPP) when both *J_sc_* and *V_oc_* decrease. Notably, *FF* cannot be used as the primary parameter representing cell performance. Increasing the interface defect density accelerates the recombination rate of charge carrier at the PTAA/absorber interface, which results in degraded device performance. Thus, a value of defect density of 1 × 10^9^ cm^−2^ could be chosen as a design parameter, yielding PCE = 22.94%.

Next, we studied the effect of defect density regarding the absorber/ETL interface (Figure 9). It has a substantial influence on the function of the solar cells, as the quality of the absorber/ETL interface exhibits a significant impact on PSCs performance. The results are shown in Figure 9. It can be observed that several parameters change very little in the defect density range from 10^9^ to 10^11^ cm^−2^, and then drop abruptly when the defect density in in a range greater than 10^11^ cm^−2^. Such a sharp decline in solar cell performance can be attributed to the increase in recombination. Therefore, in order to maintain high efficiency, the defect density should be controlled in the range of 10^9^ to 10^11^ cm^−2^. A defect density of 1 × 10^9^ cm^−2^ was chosen and, in this case, PCE = 22.94%.

## 4. Conclusions

In conclusion, we proposed the inclusion of an inorganic material (Cu_2_O) into perovskite solar cells as a buffer layer between the HTL and the electrode. We found that the insertion of Cu_2_O can greatly improve the *FF* of n-i-p devices for low work function electrodes (especially Cu), which is attributed to the fact that it can effectively lower the interfacial energy barrier between the interface and promote hole transportation. Simultaneously, it can achieve the best energy alignment among Cu, Cu_2_O, and PTAA, thereby reducing the charge accumulation, which optimizes *V_oc_* and *PCE*. Moreover, we optimized the thickness of Cu_2_O, MAPbI_3_, and PTAA to 0.02 µm, 0.60 µm, and 0.02 µm, respectively, and the appropriate defect densities at the interface between the HTL/absorber and absorber/ETL were explored to provide a reference for the experiment. The most efficient performance device we achieved showed a *PCE* of 22.94%, *J_sc_* of 24.93 mA/cm^2^, *V_oc_* of 1.19V, and *FF* of 77.04%, This simulation study has instructive implications for achieving high efficiency, as well as low-cost perovskite solar cells.

## Figures and Tables

**Figure 1 materials-15-08142-f001:**
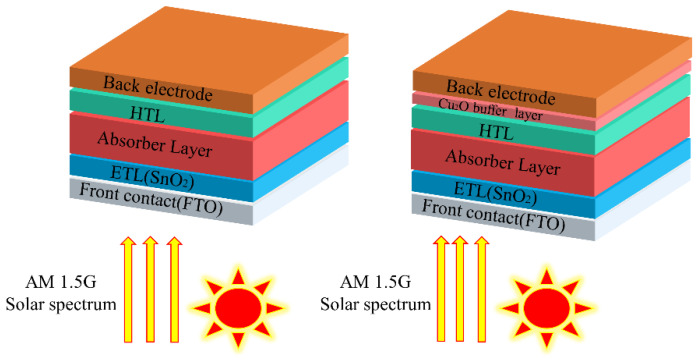
The n-i-p typical perovskite solar cell structure (**left**) and the structure with a Cu_2_O buffer layer (**right**).

**Figure 2 materials-15-08142-f002:**
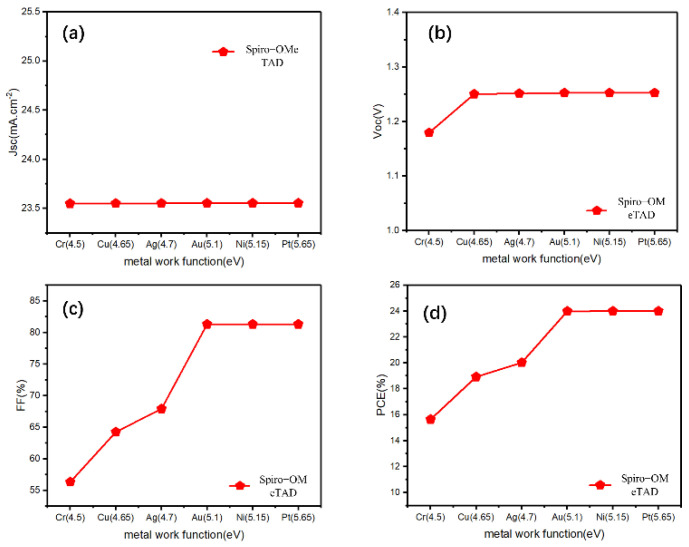
(**a**–**d**) The photovoltaic parameters (*J_sc_*, *V_oc_*, *FF*, *PCE*) for the reference n-i-p devices with Spiro-OMeTAD-HTL and different electrodes.

**Figure 3 materials-15-08142-f003:**
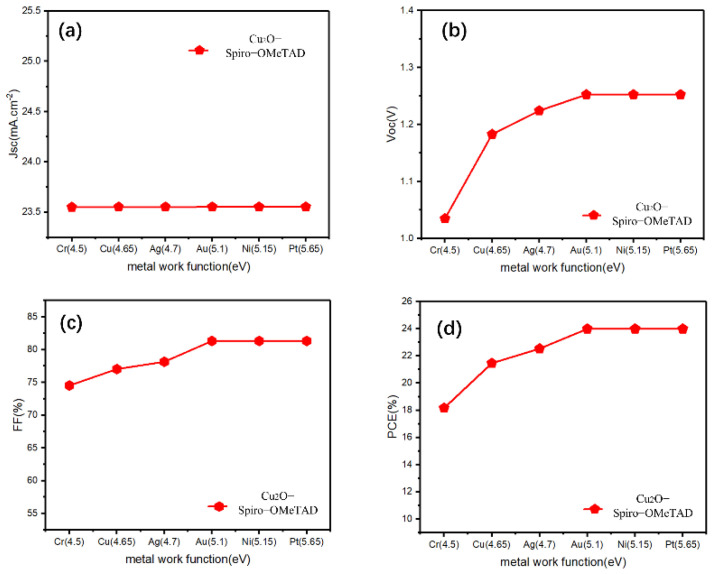
(**a**–**d**) The photovoltaic parameters (*J_sc_*, *V_oc_*, *FF*, *PCE*) for the Cu_2_O buffer layer contained devices with Spiro-OMeTAD-HTL and different electrodes.

**Figure 4 materials-15-08142-f004:**
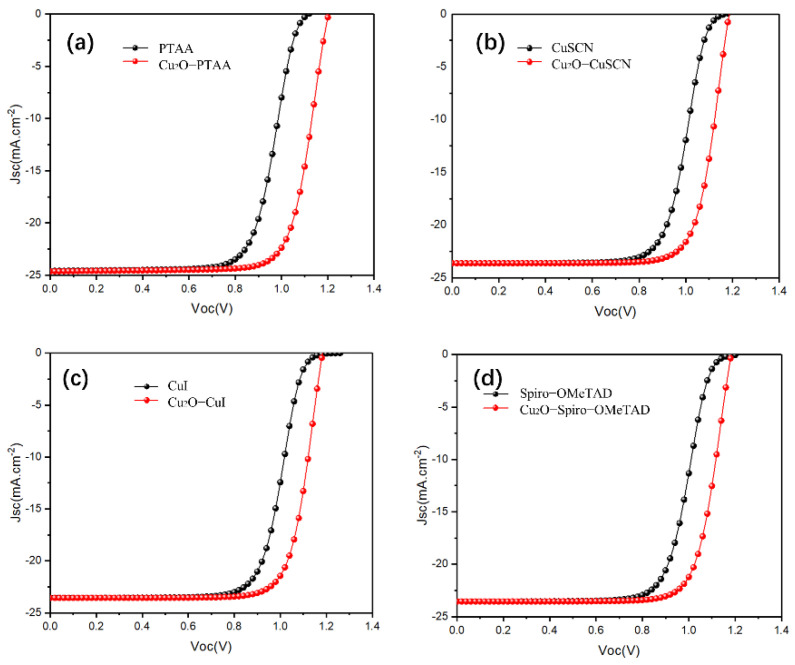
(**a**–**d**) The current density-voltage (*J–V*) characteristic curves of both device structures (with and without a Cu_2_O buffer layer) with PTAA-HTL (**a**), CuSCN-HTL (**b**), CuI-HT (**c**), Spiro-OMeTAD (**d**).

**Figure 5 materials-15-08142-f005:**
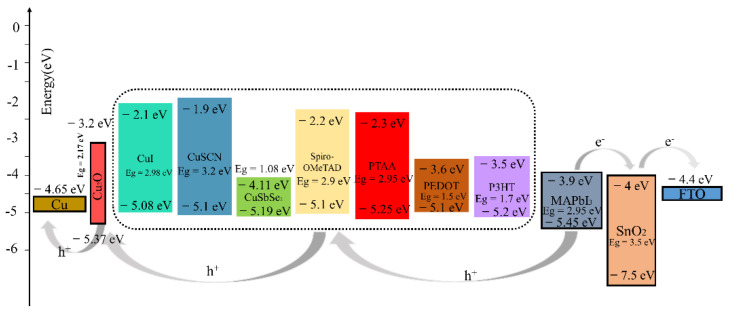
Energy diagram for devices with different HTLs.

**Figure 6 materials-15-08142-f006:**
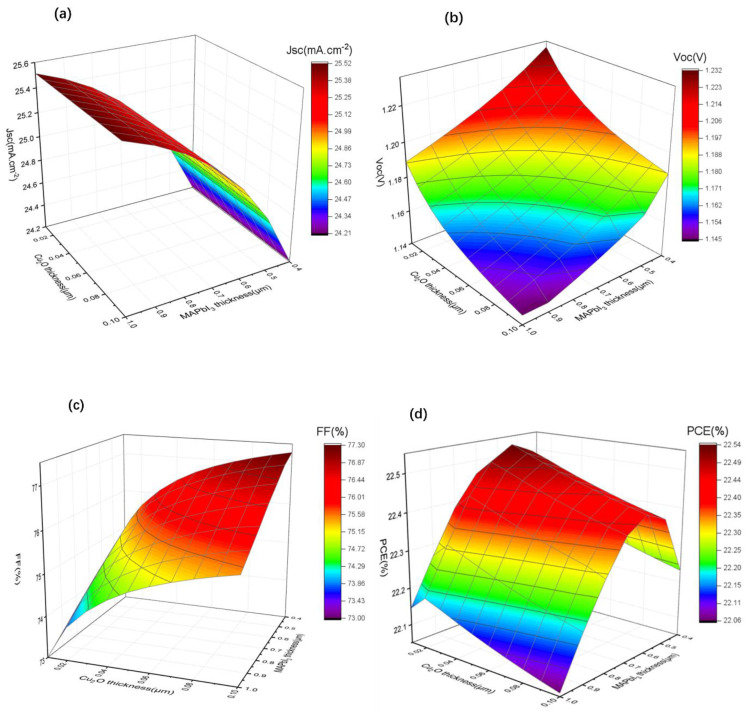
(**a**–**d**) The variation of *J_sc_*, *V_oc_*, *FF*, and *PCE* with different perovskite layer (0.4 µm to 1 µm) and Cu_2_O buffer layer (0.01 µm to 0.1 µm) thicknesses.

**Figure 7 materials-15-08142-f007:**
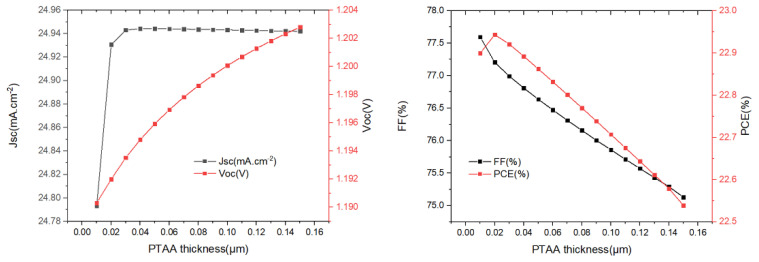
The variation of *J_sc_*, *V_oc_*, *FF*, and *PCE* with different PTAA-HTL layer (0.01 µm to 0.15 µm) thicknesses.

**Figure 8 materials-15-08142-f008:**
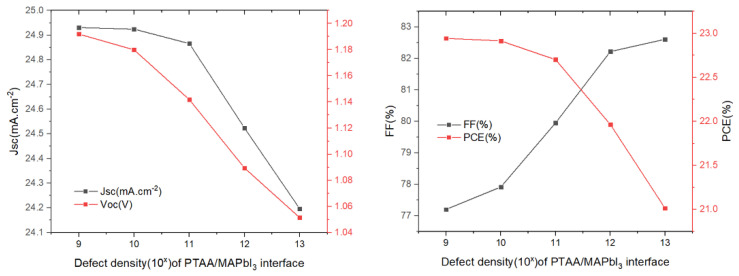
Influence of the HTL/absorber interface defect density on the photovoltaic parameters (*V_oc_*, *J_sc_*, *FF*, *and PCE*).

**Figure 9 materials-15-08142-f009:**
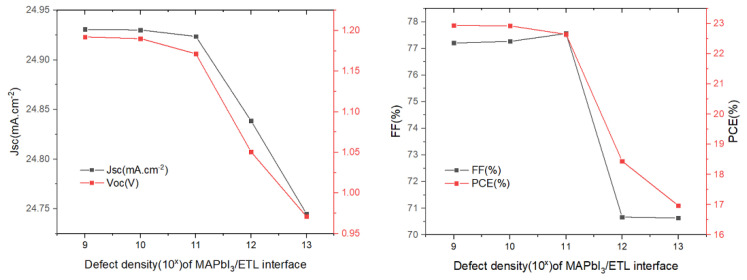
Influence of the MAPbI_3_/SnO_2_ interface defect density on the photovoltaic parameters (*V_oc_*, *J_sc_*, *FF*, *and PCE*).

**Table 1 materials-15-08142-t001:** Primary input parameters used for the simulation of perovskite solar cells.

Parameters	FTO	ETL(SnO_2_)	MAPbI_3_	HTL (Spiro-OMeTAD)	Buffer Layer (Cu_2_O)
Thickness (µm)	0.5	0.07	0.5	0.15	0.03
Band gap (eV)	3.5	3.5	1.55	2.9	2.17
E Affinity	4	4	3.9	2.2	3.2
Permittivity	9	9	30	3	7.1
Effective density of states at CB	2.2 × 10^18^	2.2 × 10^18^	2.2 × 10^18^	2.2 × 10^19^	2.5 × 10^20^
Effective density of states at VB	1.8 × 10^19^	1.8 × 10^19^	1.8 × 10^19^	2.2 × 10^19^	2.5 × 10^20^
Mobility of e^−^	20	20	2	1 × 10^−4^	200
Mobility of h^+^	10	10	2	1 × 10^−4^	8600
Density of n-type doping	2 × 10^19^	1 × 10^17^	0	0	0
Density of p-type doping	0	0	1 × 10^13^	1 × 10^18^	1 × 10^18^
Thermal velocity of e^−^	1 × 10^7^	1 × 10^7^	1 × 10^7^	1 × 10^7^	1 × 10^7^
Thermal velocity of h^+^	1 × 10^7^	1 × 10^7^	1 × 10^7^	1 × 10^7^	1 × 10^7^
Density of defects	Donor-1 × 10^15^	Donor-1 × 10^15^	Neutral-2.5 × 10^13^	Acceptor-1 × 10^15^	Acceptor-1 × 10^15^
References	[17]	[18,19]	[20]	[21]	[19,22]

**Table 2 materials-15-08142-t002:** Photovoltaic parameters obtained using different HTMs by using an Au electrode (with and without Cu_2_O).

HTL	Cu_2_O Buffer Layer	*J_sc_* (mA/cm^2^)	*V_oc_* (V)	*FF* (%)	*PCE* (%)
CuI	without	23.56	1.25	81.63	24.10
	with	23.56	1.25	81.63	24.10
	[38]	33.88	1.13	51.78	19.63
	[39]	21.89	1.27	83.12	23.14
CuSCN	without	23.63	1.25	81.73	24.23
	with	23.63	1.25	81.73	24.23
	[19]	25.62	1.19	87.81	26.74
	[38]	33.58	0.86	73.21	21.04
CuSbSe_2_	without	23.48	0.74	79.97	13.97
	with	23.72	0.76	78.66	14.16
Spiro-OMeTAD	without	23.55	1.25	81.32	23.99
	with	23.55	1.25	81.32	23.99
	[19]	25.59	1.13	81.54	23.55
	[40]	21.74	1.15	80.90	20.23
PTAA	without	24.67	1.26	80.61	25.10
	with	24.67	1.26	80.61	25.10
	[41]	41.03	0.78	74.14	23.58
PEDOT: PSS	without	26.29	0.94	83.84	20.80
	with	29.42	0.96	82.91	23.45
	[38]	28.66	1.06	62.21	18.85
	[39]	21.89	1.27	71.43	19.88
P3HT	without	23.92	1.07	81.50	20.92
	with	26.76	1.08	74.69	21.49
	[19]	25.56	0.97	86.52	21.52
	[39]	21.89	1.27	74.05	20.61

**Table 3 materials-15-08142-t003:** Input parameters for the several different HTMs.

Parameters	CuI	CuSCN	CuSbSe_2_	PTAA	PEDOT: PSS	P3HT
Thickness (µm)	0.15	0.15	0.15	0.15	0.15	0.15
Band gap (eV)	2.98	3.2	1.08	2.95	1.5	1.7
E Affinity	2.1	1.9	4.11	2.3	3.6	3.5
Permittivity	6.5	10	15	3.5	10	3
Effective density of states at CB	2.8 × 10^19^	2.5 × 10^18^	9.9 × 10^20^	2 × 10^21^	1 × 10^21^	1 × 10^22^
Effective density of states at VB	1 × 10^19^	1.8 × 10^19^	9.9 × 10^20^	2 × 10^21^	1 × 10^21^	1 × 10^22^
Thermal velocity of e^−^	1 × 10^7^	1 × 10^7^	7.3 × 10^6^	1 × 10^7^	1 × 10^7^	1 × 10^7^
Thermal velocity of h^+^	1 × 10^7^	1 × 10^7^	7.3 × 10^6^	1 × 10^7^	1 × 10^7^	1 × 10^7^
Mobility of e^−^	1.69 × 10^−4^	2 × 10^−4^	10	1 × 10^−4^	1	1.8 × 10^−3^
Mobility of h^+^	1.69 × 10^−4^	2 × 10^−4^	10	1 × 10^−4^	4	1.8 × 10^−2^
Density of n-type doping	0	0	0	0	0	0
Density of p-type doping	1 × 10^18^	1 × 10^18^	5 × 10^16^	1 × 10^18^	3.17 × 10^14^	3.17 × 10^13^
Density of defects	Acceptor-1 × 10^15^	Acceptor-1 × 10^15^	Acceptor-1 × 10^14^	Acceptor-1 × 10^15^	Acceptor-1 × 10^14^	Acceptor-1 × 10^14^
References	[38]	[19,39,47]	[48]	[21,49]	[50,51]	[19,32]

**Table 4 materials-15-08142-t004:** Photovoltaic parameters obtained using different HTLs.

HTL	Cu_2_O Buffer Layer	*J_sc_* (mA/cm^2^)	Voc (V)	FF (%)	PCE (%)
CuI	without	23.56	1.25	64.88	19.13
	with	23.56	1.18	77.40	21.58
CuSCN	without	23.62	1.17	69.29	19.10
	with	23.62	1.19	77.48	21.70
CuSbSe_2_	without	23.45	0.73	79.74	13.64
	with	23.61	0.71	69.87	11.74
Spiro-OMeTAD	without	23.55	1.22	66.07	18.92
	with	23.55	1.18	77.03	21.46
PTAA	without	24.60	1.12	69.16	18.99
	with	24.64	1.20	75.92	22.49
PEDOT: PSS	without	26.05	0.73	69.80	13.30
	with	28.14	0.90	75.82	19.27
P3HT	without	23.88	0.95	76.56	17.41
	with	25.49	1.02	72.09	18.67

## Data Availability

Not applicable.

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
