# Peer review of "The Investigation of the Influence of a Cu2O Buffer Layer on Hole Transport Layers in MAPbI3-Based Perovskite Solar Cells"

_materials, 2022, doi:10.3390/ma15228142_

Round 1

Reviewer 1 Report

The authors proposed an inorganic material (Cu2O) into perovskite solar cells as a buffer layer between hole transporting layer (HTL) and electrode. They found that the addition of Cu2O can greatly improve the FF of n-i-p devices based on low work function electrodes, leading to PCE of 22.54%, Jsc of 24.94 mA/cm2, Voc of 1.20V and FF of 75.13% by simulation. The principal idea exhibited in this manuscript is good. Thereby, I think the manuscript can be published in Materials after a minor revision. Below are the comments for the authors to address. 

1.     I find a few grammar, spelling and typing mistakes. I highly recommend that the authors should go through the text once again.

2.     The role of hole transporting layer should be mentioned obviously since this manuscript focused on that. The authors should consider following references: Advanced Energy Materials 10 (13), 1903326; Materials Science in Semiconductor Processing 108, 104908.

3.     The Cu2O buffer layer thickness is critical in this work. However, what is the concentration of Cu2O buffer layer in this simulation?

4.     The selection of different hole transporting materials should be explained.

5.     The authors stated that “the addition of Cu2O can effectively reduce the Schottky barrier between the HTL/Cu interface and facilitated hole transportation from HTL to electrode”. Please provide any references which support this statement.

6.     In Table 2, using PEDOT:PSS as HTL, the PCE of with and without Cu2O layer showed significant difference. An extended discussion is highly desirable.

7.     Since some hole transporting materials usually require additives in the real device (such as PTAA or Spiro-OMETAD), I wonder whether the presence of those additives impact on the simulation results or not. An extended discussion is highly desirable.

8.     The authors claim about the stability of the device in the manuscript. Was this work performed via simulation? I did not see that in this manuscript.

Reviewer 2 Report

This manuscript by Chunxiang Lin et al. reported Cu2O buffer layer on hole transport layer in MAPbI3 solar cells. Considering the importance of interfacial engineering, this work could attract some interest of research community in this area. However, the manuscript lacks fundamental understanding of p-type semiconductors pertaining to photovoltaic performance. The conclusion in this manuscript are not supported by the reliable results as this work has serious flaws. More detailed comments are as follows;

1.      Primary input parameters used for the simulation including thickness, band gap, and E affinity have been described in Table 1. However, parameters of metal electrode, one of the important layers, are missing. The thickness of the metal electrode is especially one of the significant factors affecting photovoltaic performance.

2.      In Table 1, the thickness of Cu2O has been written to be 30 nm. Moreover, valence band edge of Cu2O in figure 5 has been described to be -5.37 eV. Considering the function and the energy level of Cu2O as a buffer layer, hole transfer from the various HTMs to Cu electrode will be unfavourable as Cu2O buffer layer with deep valence band edge comparable to these of the applied HTMs is too thick for hole to transfer.

3.      The authours simulated the impact of Cu2O buffer layer on PCE of perovskite solar cells. However, the input parameters of amorphous or crystalline structure of the Cu2O are missing.

4.      The authours used the Cu2O p-type semiconductor as a buffer layer in this work. By any chance, have the authours applied the Cu2O as a hole transporting layer rather than a buffer layer?

5.      Cu2O used in this work is one of the two types of copper oxide. Have the authours simulated CuO, another type of copper oxide, as a buffer layer?

6.      In Table 4, input parameters of the various HTMs for simulation has been described. However, the information on amorphous or crystalline structure of the HTMs is missing.

Reviewer 3 Report

1.     From my point of view, the publication of ‘’The investigation of Cu2O buffer layer on hole transport layers in MAPbI3 based perovskite solar cells” can only be considered after major revisions.

2.     In introduction part should be improved and the author would be discussed more motivation for this work and why they use the Cu2O buffer layer.

3.     Author should improve Section 3.2. ‘’Effect of Cu2O buffer layer on various metal electrodes’’ Please explain the mechanism of performance of Cu2O. not only put the experimental value.

4.       Please explain more about the Voc increase phenomenon by using Cu2O.

Reviewer 4 Report

Manuscript ID:materials-1987502

The manuscript entitled “The investigation of Cu2O buffer layer on hole transport layers in
MAPbI3 based perovskite solar cells
” is interesting to compare different HTL ́sin order to improve the efficiency in perovskite solar cells. However, from the reviewer’s point of view, this paper could be published in this Journal (Journal of Electronic Materials) after a major revision. In this sense, some points need further clarification, as noted below.

11.-   English writing needs to be improved.

22.-  I suggest changing the manuscript name to “The investigation of Cu2O buffer layer on hole transport layers in
MAPbI3-based perovskite solar cells.

33.-  Some typos need to be corrected.

44.-  On page 6, the authors state:

…Table 2 lists the photovoltaic parameters comparison table of different HTL devices with and without Cu2O buffer layer when using Au electrode. We can find that for most HTMs, the addition of Cu2O buffer layer has little effect on the photovoltaic parameters (Jsc, Voc, FF, PCE) with Au electrode…

However, this effect is not noted for most HTLs. Please, correct this affirmation.

55.- I suggest adding simulations with other materials, such as MoO3. Cite the following work: https://doi.org/10.1007/s11082-020-02437-y

66.-    The results presented are pretty interesting. Nevertheless, from my point of view, it is necessary to add more simulations (at least HTL thickness and interfacial defect density) to publish in this journal. Please, provide it.

 7.-   Please, add the EQE results for each HTL (with and without the buffer layer)

88.-     Please, provide experimental results. Then compare them with this work.

99.-     References must be written accordingly to the journal.

Round 2

Reviewer 2 Report

The authors have adequately responded to reviewer’s comments. I recommend the manuscript for the publication in Materials.

Reviewer 3 Report

Please accept in present form.